Chemodiversity of Calophyllum inophyllum L. oil bioactive components related to their specific geographical distribution in the South Pacific region

http://orcid.org/0000-0002-1726-1771 Ginigini Joape 1
http://orcid.org/0000-0003-3331-6311 Lecellier Gaël J. 2 gael.lecellier@uvsq.fr
Nicolas Mael 3
Nour Mohammed 3
Hnawia Edouard 3
Lebouvier Nicolas 3
http://orcid.org/0000-0002-3374-2508 Herbette Gaëtan 4
Lockhart Peter 5
Raharivelomanana Phila 6
1 Pacific Natural Products Research Centre, Institute of Applied Sciences, University of the South Pacific , Suva , Fiji
2 Départment de Biologie, Université de Versailles Saint-Quentin-en-Yvelines , Versailles , France
3 ISEA EA7484, University of Caledonia , Noumea , New Caledonia
4 Spectropole, Campus de St Jérôme, Aix-Marseille Univ, CNRS, Centrale Marseille, FSCM , Marseille , France
5 School of Fundamental Sciences, Massey University , Palmerston North , New Zealand
6 EIO UMR241, Université de la Polynésie française , Faa’a, Tahiti , French Polynesia
Daehler Curtis
Electronic publication date: 2019 May 22
Publication date: 2019
Volume: 7
Electronic Location ID: e6896
Received 2018 Oct 18; Accepted 2019 Apr 2
Copyright: © 2019 Ginigini et al.
Copyright year: 2019
Copyright holder: Ginigini et al.
License: This is an open access article distributed under the terms of the Creative Commons Attribution License, which permits unrestricted use, distribution, reproduction and adaptation in any medium and for any purpose provided that it is properly attributed. For attribution, the original author(s), title, publication source (PeerJ) and either DOI or URL of the article must be cited.
License URL: https://creativecommons.org/licenses/by/4.0/

Keywords: Calophyllum inophyllum, Chemodiversity, South Pacific, Neoflavonoids, Oil, Biodiversity

Funding: AFD CPF 137901A project no 1317: No SPP46-1-2013 The research was supported by grant AFD CPF 137901A project no 1317 (No SPP46-1-2013). The funders had no role in study design, data collection and analysis, decision to publish, or preparation of the manuscript.

==============================
Background

Different parts of the tree Calophyllum inophyllum L. (nuts, leaves, roots, bark, fruits, nut oil and resin) are used as traditional medicines and cosmetics in most of the Pacific Islands. The oil efficiency as a natural cure and in traditional cosmetics has been largely described throughout the South Pacific, which led us to investigate C. inophyllum’s chemical and genetic diversity. A correlative study of the nut resin and leaf DNA from three distinct archipelagos in the South Pacific was carried out in order to identify diversity patterns in C. inophyllum across the South Pacific.

Methods

Calophyllum inophyllum plants were sampled from French Polynesia, New Caledonia and Fiji. We extracted tamanu oil (nut oil) resin for chemo-diversity studies and sampled leaf tissues for genetic studies. We applied an analysis method designed for small quantities (at a microscale level), and used High Performance Liquid Chromatography (HPLC) to establish the chemo-diversity of tamanu oil resin. In-house standards were co-eluted for qualitative determination. Genetic diversity was assessed using chloroplast barcoding markers (the Acetyl-CoA carboxylase (accD) gene and the psaA-ycf3 intergenic spacer region).

Results

Our HPLC analysis revealed 11 previously known tamanu oil constituents, with variability among plant samples. We also isolated and characterized two new neoflavonoids from tamanu oil resin namely, tamanolide E1 and E2 which are diastereoisomers. Although genetic analysis revealed low genetic variation, our multivariate analysis (PCA) of the tamanu oil resin chemical profiles revealed differentiation among geographic regions.

Conclusion

We showed here that chromatographic analysis using formalized in-house standards of oil resin compounds for co-elution studies against oil resin samples could identify patterns of variation among samples of C. inophyllum, and discriminate samples from different geographical origins.

Introduction

Chemical and medicinal properties of Alexandrian laurel (Calophyllum inophyllum), commonly known as beach mahogany, have been extensively studied throughout the world and even more in the Asia-Pacific region (Léguillier et al., 2015; Pawar, Swati & Shubhada, 2011; Patil et al., 1993). The plant has a myriad of medicinal uses, most of them involving topical applications. Recent studies such as Léguillier et al. (2015) and Ansel et al. (2016) have focused more attention on the cosmetic aspects of the plant in the Pacific region. Other common names for the plant in some Pacific island countries are dilo (Fiji), fetau (Samoa), tamanou de bord de mer (New Caledonia) and tamanu in the Cook Islands and French Polynesia (Friday & Okano, 2006). Historically, before the conversion of Polynesians to Christianity, the tamanu trees were considered sacred. They were planted inside the royal Marae (sacred areas) (Dweck & Meadows, 2002).

Calophyllum inophyllum belongs to the flowering plant family Calophyllaceae (Angiosperm Phylogeny Group, 2009; Prabakaran & Britto, 2012) and is native to the Indo-Pacific region (Africa, India, South East Asia, Australia and the Pacific islands). It grows to a height of 8–15 m and has a large canopy. The wood is widely used for making cabinet and other furniture, for carving, and for boat and canoe building. Furthermore, different parts of the plant have been used in traditional medicine and as excellent raw material for cosmetics (Dweck & Meadows, 2002). For instance, the nut oil has been used for medicine against skin infections, as a scar remover as well as for other cosmetic uses (Friday & Okano, 2006). In Fiji, the oil is used to cure arthritis and joint pain, as an eye wash for conjunctivitis (Cambie & Ash, 1994) and also to prevent infantile rash. The resin mixed with strips of bark and leaves is used as a treatment for sore eyes. The green fruit is used against tuberculosis (Cambie & Ash, 1994). In some islands of Polynesia, the oil has been used as an alternative for candle nut oil in lamps and also massaged into hair and used as a common topical application for skin diseases and burns (Prabakaran & Britto, 2012). A number of studies have revealed interesting oil biological activities such as antibacterial (Yimdjo et al., 2004), antifungal (Saravanan et al., 2011), anti-inflammatory against skin infections (Bhalla et al., 1980). Most recently it has proven useful for wound healing (Léguillier et al., 2015). Such is the uniqueness of its properties that “tamanu oil” has been recognized as an active cosmetic ingredient and recorded as C. inophyllum seed oil by the International nomenclature of cosmetic ingredients (Assouvie, 2013; Ansel et al., 2016).

The bioactive components (belonging to neoflavonoid, xanthone and triterpene secondary metabolite groups) in this plant are highly rich and recognized as having medicinal properties as shown in Table 1. The composition of the main neoflavonoid compounds in the oil are as follows: calophyllolide, inophyllums (C, D, E and P), tamanolides D and P, calanolide Gut 70, and finally the calanolides A, B and D (Laure, 2005; Leu et al., 2009; Assouvie, 2013). Also of great interest is the anti-HIV efficiency of pyranocoumarins from the Calophyllum genus (Wang et al., 2006). Recent studies have also examined their use in cicatrization, and as an anti-aging agent (Léguillier et al., 2015; Ansel et al., 2016). The calanolide A compound, a minor constituent of C. inophyllum resin extract (Ansel et al., 2016) has also attracted recent interest because it is the only natural product that has progressed into human clinical trials with positive results against HIV-1 (Wang et al., 2006).

Table 1 Summary of major bioactive compounds found in C. inophyllum resin oil with some bark and leaf constituents and their anti-cancer, anti-microbial, anti-HIV and anti-inflammatory effects.

Compounds	Anti-cancer	Anti-microbial	Anti-HIV	Anti-inflammatory	
Calophyllolide	⧾b	⧾c		⧾g	
Inophyllum P			⧾a,d		
Inophyllum C	⧾b	⧾c	⧾a		
Inophyllum D	⧾b		⧾a		
Inophyllum E	⧾b				
Inophyllum B			⧾a,d		
Inophyllum A	⧾b		⧾a		
Calanolide Gut 70	⧾h				
Calanolide A/Calanolide B			⧾d		
Calanolide/Pseudocalanolide D			⧾f		
12-Oxocalanolide			⧾d,e		
Note:

Compound References articles in parenthesis aPatil et al. (1993), bItoigawa et al. (2001), cYimdjo et al. (2004), dKostova & Mojzis (2007), eWang et al. (2006), fIshikawa (2000), gSaxena et al. (1982), hJin et al. (2011).

The chemical composition analysis of C. inophyllum leaves from various sites in French Polynesia, and that of fruits originating from various sites in India, has revealed regional differentiation (Laure, 2005; Pawar, Swati & Shubhada, 2011) between C. inophyllum plants found inland and closer to the coast. In order to evaluate the chemical qualities of the different tamanu oils of the South Pacific, phytochemical analyses are therefore necessary. Given that secondary metabolites play a role in the adaptation of plants to biotic and abiotic factors, it would be interesting to know if, as revealed by Pawar, Swati & Shubhada (2011), abiotic factors such as those associated with the geography of harvesting sites have an influence on the chemical profile of tamanu oil.

In addition to the general question of regional influence on the chemical profiles of tamanu oil, we were also interested in whether dipyranocoumrins and more particularly inophyllums can be used, as described by Pawar, Swati & Shubhada (2011), as chemotaxonomic markers for the identification of C. inophyllum from various geographical areas of the South Pacific. To address these questions, from three South Pacific countries (French Polynesia, New-Caledonia and Fiji) shown on Oceania’s map in Fig. 1. we characterized genetic diversity together with the phytochemical diversity, following a polyphasic approach, as already applied to some microbial research.

Figure 1 Map of the Oceania with the three sampled locales.

Map was downloaded from the works of Peter Fitzgerald on the Oceania Regions (https://commons.wikimedia.org/wiki/File:Oceania_regions_map.png) and the three sampled locales are indicated in added red boxes. This Figure is licensed under CC BY-SA 3.0 (https://creativecommons.org/licenses/by-sa/3.0/deed.en).

Polyphasic taxonomy is a consensus method of taxonomy developed to incorporate phenotypic, genotypic and phylogenetic data for micro-organisms (Vandamme et al., 1996). A simplified version of this method for plant research, utilizing only genotype and phenotype variation for elucidation of diversity has been successfully used by Pawar, Swati & Shubhada (2011), Lynch et al. (2016) and Hu et al. (2007). As in these previous studies, the present study is aimed at giving more insight into the diversity of the C. inophyllum plant in Fiji, French Polynesia and New Caledonia by using genetic data from barcoding universal gene markers in accD and psaA-ycf3. We expect that by studying the genetic variation and corresponding HPLC chemical profiles (phenotypic information) for each sample, we can gain new insights that could help explain current diversity patterns of this plant. Our approach differs from Pawar, Swati & Shubhada (2011) in that it does not utilize simple sequence repeat markers. Instead, we used sequence data from two chloroplast regions that have been employed for barcoding: the accD gene and the psaA-ycf3 intergenic spacer region. The plastid accD gene, which encodes for the ß-carboxyl transferase subunit of acetyl co-enzyme A carboxylase, is present in the plastids of most flowering plants, including non-photosynthetic parasitic plants and is involved in fatty acid biosynthesis. Associated with a fast evolving genome region in some evolutionary lineages, it has been used in barcoding experiment analyses for Magnoliophyta (Lahaye et al., 2007), Mesangiospermae (Lam, Merckx & Graham, 2016) and Fabids (Xi et al., 2012). Studies with Chlamydomonas reinhardtii and higher plants have shown that ycf3 is required for the assembly of the Photosystem 1 complex (Boudreau et al., 1997; Ruf, Kossel & Bock, 1997).

Materials and Methods

Sample collection

Tamanu nuts were collected during the fruit flushing season from June to August and even collected late in December. In New Caledonia, tamanu nuts were collected under the scientific authorization of the South Province No 2050-2014. In Fiji, all samples were collected along roadsides at each collection site. At each location, a replicate sampling plan was applied as observed in the geodata (Table 2). A total of two leaf samples were collected at each tree together with three to five nuts. From the leaf composite, one leaf was placed in a ziplock bag as a voucher and two leaf samples were stored at −80 °C or immediately placed into DNA/RNA Shield™ (Zymo Research, Irvine, CA, USA) solution for DNA extraction. The nuts were placed in polyethylene trays and aired to remove moisture for 2 weeks before they were placed into a solar drier and dried for 6–8 weeks. In total, there were 22 sampled sites and a total of 85 trees sampled from the three major collections in Fiji, French Polynesia and New Caledonia. Note that due to the nature of collections being conducted in isolated locations from both French Polynesia and New Caledonia, limited amount of material was collected and thus vouchers specimens could not be deposited for these two research groups. Only the samples collected in Fiji had vouchers deposited at the South Pacific Regional Herbarium (Table S1).

Table 2 Sampling plan with GPS and DNA sample pretreatment for all samples.

Country	Island	GPS coordinates	Position/village (sample label)	Nuts	Replicates/site	
Fiji	Viti Levu	18°18′19″S 178°33′38″E	South (F-S1-5)	5	5	
18°18′52″S 177°34′21″E	West (F-W1-5)	5	5	
17°53′56″S 178°44′44″E	East (F-E1-5)	5	5	
17°38′30″S 170°35′47″E	North (F-N1-5)	5	5	
Rotuma	12°30′0″S 177°4′59″E	(R1-R5)	5	5	
New Caledonia	Main Iand	22°18′05″S 166°26′42″E	Nouméa (N1-3)	4	3	
22°09′03″S 166°56′02″E	Yaté (N4-6)	5	3	
French Polynesia	Rurutu	22°25′60″S 151°19′59″W	N.D. (T1-T3)	3	3	
Nuku Hiva	8°54′0″S 140°6′0″W	Taiohae (T4-T6)	1	3	
Tahuata	9°57′0″S 139°4′59″W	Apatoni (T7-T9)	3	3	
Moorea	17°31′60″S 149°49′59″W	Paopao (T10-T12)	3	3	
		Haapiti (T13-T15)	1	3	
		Taevaro (T16-T18)	1	3	
		Opunohu (T19-T21)	1	3	
Raiatea	16°49′60″S 151°25′1″W	Avera (T22-T24)	3		
Tahaa	16°37′60″S 151°30′0″W	Patio (T25-T27)	3	3	
Tahiti	17°30′0″S 149°30′0″W	Hitiaa (T28-T30)	3	3	
		Paea (T31-T33)	3	3	
Anaa	17°25′0″S 145°30′0″W	N.D. (T34-T36)	2	3	
Apataki	17°25′0″S 145°30′0″W	N.D. (T37-T39)	2	3	
Fakarava	16°19′60″S 145°37′1″W	Rotoava (T40-T42)	3	3	
Rangiroa	15°10′0″S 147°34′59″W	Tiputa (T43-T45)	7	3	
Raroia	16°1′0″S 142°27′0″W	Ouest (T46-T48)	3	3	
Takapoto	14°37′60″S 145°12′0″W	Fakatopater (T49-51)	3	3	
Tikehau	15°0′0″S 148°10′1″W	Tuherahera (T52-54)	2	3	

Microscale extraction, purification and HPLC analysis

From each of the 85 collected trees, a set of three to five nuts weighing 7–10 g were picked up and subjected to small scale cold extraction consisting of maceration and nut grinding in a cheese cloth followed by crushing in a mortar and pestle to allow oil expression through the cloth. Samples were washed with EtOAc (provided by VWR Chemicals, Radnor, PA, USA) and sonicated for 5 min to extract all nut components. These were dried in vacuo to yield a crude extract containing classical fatty acids components and de-fatted compounds. This last crude extract (oil) was then partitioned with EtOH (provided by Fisher chemical, Hampton, NH, USA) at a 1:1 ratio v/v to extract only EtOH soluble non fatty compounds. The EtOH layer was removed and dried under vacuum yielding a resinous extract (called resin) containing neoflavonoids compounds, which will be submitted to further chemical analysis. Any remaining oil was de-fatted twice with 40 mL hexane in total. Resinous extracts were dissolved in EtOAc: =cyclohexane (provided by VWR Chemicals) at a 1:1 ratio v/v at a concentration of 10 mg/mL for final injection. Chromatographic analyses were performed using an Agilent 1100 series gradient HPLC fitted with a UV/DAD system. The HPLC column used was an Interchrome Modula-Cart QS Uptisphere five µm Si column, and the data were viewed on Agilent Chemstation software (Santa Clara, CA, USA). Optimized step gradient elution was applied utilizing Cyclohexane (HPLC grade from Fisher chemical) and EtOAc (HPLC grade from Fisher chemical) as solvent and the analytical conditions were as follows: flow rate one mL/min, pressure 2,500 psi max, injection volume of four µL. The solvent system gradient conditions are shown in Table S2 in the supporting information.

HPLC analysis of the samples from all three locations (Fiji: RawDataS1, New Caledonia: RawDataS2 and French Polynesia: RawDataS3) involved numerous trials to obtain the maximum amount of compounds in the shortest runtime. On completion of the analysis, 80 resin samples were chosen from the total of 85 samples and five were discarded due to poor alignment and high signal to noise. Two more samples were discarded due to poor alignment. In total, 78 samples were analyzed by HPLC. Sample chromatogram profiles were recorded at 280 nm, wavelength at which most peaks were observed in the optimized analytical conditions for neoflavonoid standards. The removal of noise, redefinition and alignment of all data was necessary to identify and assign each sample peak before comparing these to standard compounds which were isolated and identified in previous studies (Leu et al., 2009; Leu, 2009; Laure, 2005; Laure et al., 2008). Background fitting and identification of major peaks of the raw HPLC data were performed using the R package align DE v2.0.1. Chromatograms were aligned using a procedure in Scilab version 5.5.1 (Scilab Enterprises, 2012) derived from chromaligner (Wang et al., 2010).

Purification of known compounds from commercial tamanu oil resin and isolation of new compounds

A batch of oil resin extract (157 g) from commercial tamanu oil (provided by “Laboratoire de Cosmétologie du Pacifique Sud” manufacture) from French Polynesia was first partitioned with EtOH (provided by Fisher chemical) and an aqueous alkaline solution of Na2CO3 (from VWR chemicals) (10%, v/v). Its organic fraction was washed with distilled water and then dried with MgSO4 (provided by VWR chemicals) to give a neutral fraction (53 g) after solvent evaporation. This fraction was submitted to flash liquid chromatography on an open column with a silica gel (240–300 mesh) using a stepwise gradient from cyclohexane (provided by VWR chemicals) to EtOAc (provided by VWR chemicals), yielding 12 fractions. Fractions having similar Rf values on silica gel TLC (cyclohexane-acetone, both provided by VWR chemicals, 60:40, v/v) were combined. Fractions 7, 9 and 11 were submitted to repeated preparative HPLC using a Varian Dynamax Si column (250 × 21.4 mm id, five μm with cyclohexane-EtOAc (10:90) in isocratic eluent conditions. This chromatographic purification network led to the isolation of new compounds tamanolides E1 and E2 as a mixture (two mg) besides standard known compounds namely, calophyllolide, inophyllums (C, D, E, P), calanolides (Gut 70 and A, 12-oxo-calanolide) and tamanolides (D, P).

Analysis of new compounds

New compounds were identified by NMR and were further analyzed with a Bruker Avance DRX500 spectrometer (1H—500.13 MHz) equipped with a five mm triple resonance inverse Cryoprobe TXI (1H/13C/15N) in CDCl3-99.8%, (δ1H = 7.26 pm, δ13C = 77.16 ppm). The HR-ESI-MS data were collected using a QStar Elite mass spectrometer (Applied Biosystems SCIEX, Concord, ON, Canada) equipped with an ESI source operated in positive ion mode. Optical rotations were measured with a Perkin-Elmer 241 polarimeter equipped with a sodium (589 nm) lamp and a one dm cell. The FTIR spectra were established with a Thermo-Nicolet IR 200 spectrometer (Waltham, MA, USA) on a KBr cell and four cm−1 resolution.

DNA extraction and amplification

Plant DNA was extracted from young leaf tissue using the Plant Nucleospin© II kit (Macherey-Nagel) according to the manufacturer’s instruction. The leaf samples were first lyophilized in liquid nitrogen and crushed before being subjected to the manufacturer’s protocols. Two plant universal chloroplast regions were targeted for amplification namely; the Acetyl-CoA carboxylase (accD) gene and the psaA-ycf3 intergenic spacer region. The amplifications were performed in a Thermo Scientific Arktik Thermo Cycler. The PCR was performed using a Qiagen Taq polymerase kit and primers were diluted to a concentration of 0.6 µM. All sample amplifications were eluted in distilled milliQ water at 50 µL volumes and tested for yield with 1% agarose gel electrophoresis. The PCR program consisted of 92 °C of incubation during 2 min for one cycle and eight cycles of PCR (denaturation at 92 °C for 30 s followed by specific annealing temperatures for accD (44 °C) and psaA-ycf3 (43 °C) all at 30 s, and 72 °C for 1 min), followed by 40 cycles of (92 °C for 30 s, extension at primer specific temperature’s accD (46 °C) and psaA-ycf3 at 30 s, 72 °C for 1 min). A final 5 min extension was then carried out at 72 °C and for only one cycle.

Sequencing alignment and data analysis

The samples were sequenced by GATC Biotech sequencing services, Germany. Data received in AB1 format were viewed under the Applied Biosystem’s sequencing scanner and corrected for contiguous read lengths and miscalled sequence data were removed especially in the beginning and at the ends of the raw data files. Files were then aligned using MEGA6 (Tamura et al., 2013) before they were matched to nucleotide sequences using the BLASTn tool available at https://blast.ncbi.nlm.nih.gov/Blast.cgi. Only matches with 96–100% similarity were included in the construction of a multiple sequence alignment. Sequences were aligned using ClustalW (Thompson, Higgins & Gibson, 1994) and trees were generated using the MEGA6 phylogeny tools. New haplotypes were deposited into the NCBI Genbank database using the Bankit tool. Accession numbers are given in Table S3.

Phylogenetic construction

All newly determined sequences were checked using the sequencer software for peak intensity and also contiguous length. All sequences with contiguous lengths greater than 200 bp were considered for tree construction using MEGA6. Maximum likelihood (ML) trees were constructed for all accD sequences which satisfied our contiguous length criterion. Initial tree(s) for a heuristic search were first obtained by applying Neighbor-Joining (Saitou & Nei, 1987) and BioNJ (Gascuel, 1997) algorithms to a matrix of pairwise distances estimated using the maximum composite likelihood approach implemented in MEGA, and tree searches were conducted assuming a Tamura-Nei model and a discrete gamma distribution to model evolutionary rate differences among sites (five categories (+G, alpha parameter = 0.1523)). The analysis involved 18 nucleotide sequences and all positions containing gaps and missing data were eliminated before tree building. There was a total of 234 positions in the final dataset. The optimal tree obtained has been drawn to scale, with branch lengths indicating the number of substitutions per site. A ML tree was also constructed for the psaA-ycf3 Fiji sequence data only, as the Tahiti and New Caledonia DNA for this chloroplast region could not be amplified, possibly due to DNA deterioration. A Tamura-Nei model (Tamura & Nei, 1993) was assumed. The optimal tree has been shown, drawn to scale, with branch lengths determined by the number of substitutions per site. Non parametric bootstrap values are shown for internal branches. The analysis of this second data set involved 11 nucleotide sequences. Positions containing gaps and missing data were eliminated. There was a total of 55 positions in this dataset.

Statistical analysis

Multivariate statistical analysis was applied to all the chemical data in order to reveal the extent to which the 12 major compounds making up the chemical compositions of the 47 C. inophyllum oil resin samples were geographically distributed. Data were normalized by log transformation prior to principle component analysis (PCA). PCA was performed with the package Ade4 using R version 3.1 software (R Development Core Team, 2018) and a PCA biplot was drawn using Microsoft Excel software. The diversity of chemical compounds that contributed to the highest discrimination at a geospatial level were visualized and a scatter plot was generated from Microsoft Excel.

Results

Chemical composition

The fractionation, purification and identification of resin extracts (by spectroscopic methods) yielded 11 compounds from commercial tamanu oil (Figs. 2A–2I), which were identified as constituents previously reported (Laure et al., 2008; Leu et al., 2009). In addition to calophyllolide, inophyllums (C, D, E, P), calanolides (Gut 70 and A, 12-oxo-calanolide) and tamanolides (D, P), these analyses also led to the isolation of two new compounds X1 and X2 as an epimeric mixture. The presence of these two compounds X1 and X2 from the same peak in two fractions was clearly shown in 1H theoretical (Fig. 3A) and experimental spectra (Fig. 3B). As the chemical shifts shown in Table 3 of these compounds were quite similar, evidence of their epimeric existence was indicated by 1H NMR revealing more signals than expected from one compound. These most likely correspond to two sets of signals for two very close compounds that we propose here to be X1: tamanolide E1 and X2: tamanolide E2 (Fig. 2J). The chemical characteristics of tamanolide E1 and E2 are: amorphous yellowish powder; [α]25D=−19.6 (c 0.003, CHCl3). FTIR (CCl4): 3,075, 2,979, 2,926, 2,908, 2,877, 2,832, 1,740, 1,695, 1,643, 1,606, 1,575, 1,461, 1,382, 1,209, 1,152, 1,121 cm−1. 1H NMR (CDCl3, 500 MHz) and 13C (CDCl3, 125 MHz) (see Table 3); HR-ESI-MS m/z 383.1850 [M + H]+ (calcd. for C23H27O5, 383.1853).

Figure 2 Structures of isolated compounds.

(A) Calophyllolide, (B) Calanolide GUT 70, (C) Tamanolide, (D) Inophyllums P, D and C, (E) Tamanolide P, (F) 12-Oxo-calanolide, (G) Calanolide A, (H) Inophyllum E, (I) Tamanolide D and (J) Tamanolide E1,E2.

Figure 3 The superimposition of two NMR spectra’s of the mixture of tamanolide E1 and E2 at different proportions (40/60 and 70/30).

(A) Experimental (black) and theoretical (E1: red and E2: blue) 1H NMR and (B) spectra CDCl3—500 MHz.

Table 3 NMR data for tamanolide E1, tamanolide E2, calanolide D, inophyllum E at 500 MHz (CDCl3, 300 K).

Atom	Tamanolide-E1 (diast-1/2 70/30)	Tamanolide-E2 (diast 1/2 40/60)	Tamanolide D	Inophyllum-E	
δ13C	δ1Η	δ13C	δ1Η	δ1Η	δ1Η	
2	160.4 (C)	–	160.4 (C)	–	–	–	
3	109.3 (CH)	6.16 (brs)	109.3 (CH)	6.16 (brs)	6.08 (brs)	6.05 (s)	
4	163.0 (C)	–	163.0 (C)	–	–	–	
4a	104.7 (C)	–	104.7 (C)	–	–	–	
4b	156.2 (C)	–	156.2 (C)	–	–	–	
6	79.4 (C)	–	79.4 (C)	–	–	–	
7	127.2 (CH)	5.60 (d, 10.0)	127.2 (CH)	5.60 (d, 10.0)	5.54 (d, 10.0)	5.41 (d, 10.0)	
8	116.1 (CH)	6.66 (d, 10.0)	116.1 (CH)	6.66 (d, 10.0)	6.66 (d, 10.0)	6.55 (d, 10.0)	
8a	105.9 (C)	–	105.9 (C)	–	–	–	
8b	158.9 (C)	–	158.9 (C)	–	–	–	
10	77.4 (CH)	4.69 (qd, 6.6, 3.4)	77.4 (CH)	4.70 (qd, 6.6, 3.4)	4.51 (qd, 6.6, 1.8)	4.72 (qd, 6.6, 3.4)	
11	46.2 (CH)	2.68 (qd, 7.2,3.4)	46.1 (CH)	2.69 (qd, 7.2, 3.4)	2.03 (qdd, 7.2, 2.1; 1.8)	2.71 (qd, 7.2, 3.4)	
12	191.7 (C)	–	191.6 (C)	–	–	–	
12a	103.2 (C)	–	103.2 (C)	–	–	–	
12b	155.9 (C)	–	155.9 (C)	–	–	–	
13	37.5 (CH)	3.79 (brsxt, 7.0)	37.5 (CH)	3.79 (brsxt, 7.0)	3.86 (brsxt; 6.6)	–	
14	29.5 (CH2)	1.75 (m)	29.6 (CH2)	1.75 (m)	1.78 (m)	7.37 (m)	
1.45 (m)		1.45 (m)	1.46 (m)		
15	12.0 (CH3)	0.95 (t, 7.4)	12.0 (CH3)	0.96 (t, 7.4)	0.97 (t, 7.3)	7.22 (m)	
16	20.1 (CH3)	1.22 (d, 6.7)	20.0 (CH3)	1.22 (d, 6.7)	1.22 (d, 6.9)	7.37 (m)	
17	–	–	–	–	–	7.22 (m)	
18	–	–	–	–	–	7.37 (m)	
19	28.3 (CH3)	1.54 (s)	28.3 (CH3)	1.53 (s)	1.48 (s)	0.97 (s)	
20	28.2 (CH3)	1.53 (s)	28.2 (CH3)	1.53 (s)	1.49 (s)	0.95 (s)	
21	16.2 (CH3)	1.42 (d, 6.6)	16.2 (CH3)	1.41 (d, 6.6)	1.43 (d, 6.6)	1.42 (d, 6.6)	
22	9.4 (CH3)	1.15 (d, 7.2)	9.4 (CH3)	1.15 (d, 7.2)	0.80 (d, 7.2)	1.18 (d, 7.3)	
Note:

δ in ppm, (13C multiplicity determinate by 13C-DEPTQ135 and HSQCed), (br, broad; s, singlet; d, doublet; t, triplet; q,quadruplet; sxt, sextuplet; m, multiplet; J Hz).

Chemical diversity of C. inophyllum within the three regions

HPLC analysis of C. inophyllum was performed to investigate the phytochemical components of tamanu oil in different samples and their geographical distribution. A total of 11 isolated compounds (Fig. 4: peaks 1–11), along with two new compounds (Fig. 4: peak 12), were used as external standards to qualitatively analyze four sample of C. inophyllum from three different locations, Fiji West (FW1), New Caledonia Noumea (N1 and N3) and Rotuma Fiji (R4). The phytochemical components of each sample were identified by comparing the retention time of the external standards (Fig. 4).

Figure 4 Four representative HPLC chromatograms superimposed at λ 280 nm UV range detection.

The standard peaks, (1) calophyllolide, (2) tamanolide, (3) calanolide Gut 70, (4) inophyllum D, (5) tamanolide D, (6) calanolide A, (7) inophyllum P, (8) tamanolide P, (9) inophyllum C, (10) 12-oxo-calanolide, (11) inophyllum E, (12) tamanolide E1/E2, have been assigned in the chromatogram for Fiji West FW1 (orange), Rotuma Fiji sample R4 (light blue) and two New Caledonia Noumea samples N3 (dark red) and N1 (dark blue).

The alignment of all samples through R and Scilab provided a matrix containing the integrated peak area of the identified features. Data were normalized by log transformation prior to PCA analysis. The samples can be discriminated into three main regions (Fig. 5) with the French Polynesian population on the left of the graph, the New Caledonia population on the bottom right and the Fiji population on the upper right. Within French Polynesia and Fiji, no discrimination between the sites or archipelagos was observed with these first two components.

Figure 5 The PCA Scatter plot revealing the chemical diversity of C. inophyllum within the three study areas.

The first two components correspond to 15.81% and 7.59% of the total variance respectively. Orange circles for Fiji, green squares for New Caledonia and blue diamonds for French Polynesia.

The different proportions of the compounds leading to biogeographic discrimination of the three regions is shown in the variables factor map (Fig. 6). The first component, representing 15.8% of the total variance, discriminated notably tamanolide E1/E2 and Gut 70 calanolide from inophyllum C and tamanolides D and P. The proportion of tamanolide E1/E2, Gut 70 calanolide, tamanolide and inophyllum E were the main and significant compounds prominent in the composition of the oil resin from French Polynesia, while tamanolide P, D and inophyllum C were more prominently represented in the Fiji samples. The second component (7.6% of the total variance) discriminated mainly unknown compounds and calophyllolide, compound P73 positively and calophyllolide, P45, P50 and P69 negatively. Characteristic of the oil resin from New Caledonia were constituents such as calophyllolide and compound P69, which although unknown and unidentified, appeared to be unique in its occurrence as suggested by the PCA analysis. The chemical diversity of the Tahiti samples namely from the Tuamotu’s and Australes as seen in Fig. 5, were also unique with the presence of the compound P73.

Figure 6 Variables factor map.

PCA biplot generated using HPLC peak intensity variable’s data. Uncharacterized compounds are mentioned Px.

Genetic discrimination

The best fitting models of evolution determined by the MEGA6 option were respectively Tamura 3 model and Tamura-Nei model as substitution models for accD marker gene and psaA-Ycf3 spacer region. Phylogenetic relationships assuming this model of substitution for C. inophyllum are shown in Figs. 7A and 7B for the accD gene and for the psaA-ycf3 spacer region, respectively. Interestingly, only Raiatea from French Polynesia showed a unique genotype, where it produced two new haplotypes (T22 and T23) in the accD tree caused by substitutions at positions 134 and 146 bp.

Figure 7 Maximum likelihood trees for combined tamanu samples.

(A) using the accD marker gene constructed using the Tamura 3 model and (B) depicts the Fiji (PsaA-Ycf3 spacer region) using the Tamura-Nei model. Both trees are supported by 500 bootstrap iterations. The scale represents the number of substitutions per site.

Discussion

New chemical isolates and chemical variability

Compounds X1 and X2 had a molecular formula of C23H27O5 as determined by high-resolution mass spectrum (HR-ESI-MS) (m/z 383.1850 [M + H]+, calcd 383.1853) and NMR data implying 11 degrees of unsaturation (Table 3; Figs. S1–S10). The 13C NMR spectrum gave a total of 23 separated resonances and the 135-DEPTQ sequence showed the presence of six methyl, one methylene, six methine and 10 quaternary carbons including a ketone carbonyl at δ 191.7 ppm for compounds X1 and X2. The 1H resonances of both compounds (Figs. S1 and S9) were typical of a pyranocoumarin as characterized by the downfield shifted proton at δ 6.16 (1H, s, H-3) for an α,β-unsaturated lactone (ring B), and a set of doublet signals (δ 6.66 ppm (1H, d, J = 10.0 Hz, H-8), 5.60 ppm (1H, d, J = 10.0 Hz, H-7)), and two methyl singlets at δ 1.54 ppm (3H, s, H-19) and δ 1.54 (3H, s, H-20) for the first one (X1) and at δ 1.53 ppm (3H, s, H-19) and δ 1.53 ppm (3H, s, H-20) for the second one (X2), for a pyrane ring (ring C).

Furthermore, the cycle ring D showed a characteristic cis-configuration with a coupling constant of 3.4 Hz between H-10 δ 4.69 ppm (1H, qd, J = 6.5, 3.4 Hz) and H-11 δ 2.68 ppm (1H, qd, J = 7.2, 3.4 Hz) for X1 and between H-10 δ 4.70 ppm (1H, qd, J = 6.5, 3.4 Hz) and H-11 δ 2.69 ppm (1H, qd, J = 7.2, 3.4 Hz) for X2 and by NOE crosspeaks correlation between the two methyl groups H-21 δ 1.42 ppm (3H, d, J = 6.6 Hz), H-22 δ 1.15 ppm (3H, d, J = 7.2 Hz) for X1, and H-21 δ 1.41 ppm (3H, d, J = 6.6 Hz), H-22 δ 1.15 ppm (3H, d, J = 7.2Hz) for X2, like inophyllum E.

With the aid of the COSY experiment, an isobutyl unit was identified by further analysis of the remaining 1H resonances (δ 3.79 ppm (1H, brsxt, J = 7.0 Hz, H-13), 1.75 ppm (1H, m, H-14a), 1.45 ppm (1H, m, H-14b), 0.95 ppm (3H, t, J = 7.4 Hz, H-15) and 1.22 ppm (3H, d, J = 6.7 Hz, H-16)) for X1 and δ 3.79 ppm (1H, brsxt, J = 7.0 Hz, H-13), 1.75 ppm (1H, m, H-14a), 1.45 ppm (1H, m, H-14b), 0.96 (3H, t, J = 7.4 Hz, H-15) and 1.22 ppm (3H, d, J = 6.7 Hz, H-16)) for X2. This isobutyl moiety was assigned to be at the C-4 position, based on the HMBC crosspeaks between H-3 (δ 6.16 ppm) and C-13 (δ 37.5 ppm), between H-16 (δ 1.22 ppm) and C-4 (δ 104.7 ppm), between H-14 (δ 1.45, 1.75 ppm) and C-4 (δ 104.7 ppm), between H-13 (δ 3.79 ppm) and the carbons C-3 (δ 109.3 ppm) and C-4a (δ 104.7 ppm). The rest of the HMBC correlations used to obtain a complete assignment of the 1H and 13C NMR chemical shift, have been summarized and presented in Table 3. The 1H and 13C NMR of both compounds were almost identical to those of inophyllum E with the exception of an isobutyl group at C-4 and tamanolide D with the exception of the ring D.

Complete assignments of X1 and X2 were made based on 1D and 2D NMR experiments, of which compound X1 was suggested as tamanolide E1 while compound X2 was the H‐13 epimer of X1 and suggested as tamanolide E2. Proton NMR spectrum (Fig. 3) of different HPLC fractions revealed X1 and X2 as a mixture from two compounds whose signals were close to each other with a varied ratio (from 70/30 to 40/60). The chemical shifts of both compounds are listed in Table 3. Different fractions of a collected HLPC peak show a variation of proportion related to these two epimer compounds X1 and X2 and corresponding to two close sets of signals in the 1H NMR spectrum (Fig. 3). Assignment of the resonances was based on the 1D and 2D NMR experiments (1H, 13C-DEPTQ135, COSY, HSQCed, HMBC and NOESY).

Finally, the structures of 11 known compounds are shown (Fig. 2) including the two new compounds assigned as tamanolide E1 and E2 after elucidation from exhaustive NMR analysis (1H, 13C-DEPTQ135, COSY, HSQCed, HMBC, NOESY), including the comparison of experimental and theoretical results (Figs. 3A and 3B) and by comparison with NMR data of the phenylcoumarin back bone from inophyllum E and tamanolide D.

Consistent with earlier findings (Prabakaran & Britto, 2012; Laure et al., 2008), calophyllolide remains as the dominant major compound in tamanu resin for all isolated chemotypes. Calanolide A which is the most potent of all C. inophyllum compounds with anti-HIV-1 activity (Wang et al., 2006) was found as a minor component in C. inophyllum oil. The pyranocoumarines inophyllum B and P (also active against HIV-1) were also isolated. The former is not only the most active but is also the only natural product undergoing clinical trials against HIV-1 (Wang et al., 2006). As seen in Fig. 4, the 280 nm UV spectrum revealed the maximum number of peaks and therefore standards and samples were targeted for visualization at this wavelength.

All the standards isolated here as seen in Fig. 4 have been isolated in previous studies from French Polynesian C. inophyllum oil (Leu, 2009; Leu et al., 2009; Laure, 2005). A unique characteristic of C. inophyllum oil is the presence of a resin which has been shown to contain neoflavonoid biologically active constituents (Ansel et al., 2016). Previous studies by McKee et al. (1998) and Li et al. (2007) have indicated the use of HPLC analyses to investigate differences in chemical profiles from different populations. In our case, we have looked at samples from three localities. The population from New Caledonia was characterized by peaks not found in earlier studied populations and these novel peaks suggest new and specific chemomarkers. Unique to this study as shown in the biplot on Fig. 6 is that we were able to show the chemodiversity of tamanu resin oil across the three study sites based on chemical compounds that contribute to the highest discrimination at a geographical level from the PCA analysis. This point is important and rather unique in the study as this reveals the chemical specificity of compounds and their proportions in tamanu resin by geographical locations across the South Pacific.

DNA variability

This is the first reported work on C. inophyllum using the universal chloroplast barcoding regions the gene accD and the intergenic region psaA-ycf3. Low levels of genetic variation were found with these markers, and this variation was not functionally linked to differences in chemical composition. The absence of genetic variation among Fiji psaA-ycf3 regions suggests low genetic variation in C. inophyllum. However, analyses of additional chloroplast gene regions or nuclear ISSR markers may yet reveal more discrimination within the Fiji archipelago and between populations from different archipelagos.

Conclusions

Our results are informative in revealing that chemical differences in tamanu resin can be a tool for the discrimination of samples and geographic regions. In our case, chromatographic data proved to be more informative and highly discriminative than DNA barcoding data, possibly owing to low genetic variation in the used chloroplast regions. Additional chloroplast barcoding regions or utilization of nuclear microsatellites may give a better perspective on patterns of genetic diversity. Of interest were haplotypes (T22 and T23) which were distinct from the more commonly found C. inophyllum genotype. Polyphasic taxonomy that considers both chemical diversity and genetic diversity presents the best approach to delineate biological variation across geographical boundaries. However, higher levels of genetic resolution are required to characterize variation in the disjunct distribution of C. inophyllum across the South Pacific and to bring insight into the diversification processes that have occurred following geographic isolation. This diversification includes the evolution of novel pyranocoumarins compounds characterized by this study: tamanolide E1 and E2 (C-13 epimers as a mixture).

Supplemental Information

Supplemental Information 1 Supporting Information.

Supplementary tables (S1–S3) and figures (S1–S10).

Click here for additional data file.

Supplemental Information 2 Raw data HPLC Chromatograms from Fiji.

Time and absorbance values at 280 nm. The site and the number of the collected sample from Fiji are in the separate file names.

Click here for additional data file.

Supplemental Information 3 Raw data HPLC Chromatograms from New Caledonia.

Time and absorbance values at 280 nm. The site and the number of the collected sample from New Caledonia are in the separate file names.

Click here for additional data file.

Supplemental Information 4 Raw data HPLC Chromatograms from French Polynesia.

Time and absorbance values at 280 nm. The site and the number of the collected sample from French Polynesia are in the separate file names.

Click here for additional data file.

This paper is dedicated to the memory of our wonderful colleague, supervisor and friend Prof William Aalbersberg who has passed away. We thank and greatly appreciate him for his wisdom and inspiring research leadership. We are grateful to the people of French Polynesia, Fiji and New Caledonia for giving us access to sampling areas to collect our nuts and leaf samples. We thank Cloe Check, Juliette Prevost, Nicolas Martin, and Ranitea Ly for performing HPLC analysis on all the samples. We also thank Mr. John Bennett for supplying oil extracts and assisting in the processing of samples in Fiji as well as Mr. Olivier Touboul (LCPS) for tamanu sample collection in French Polynesia. Special thanks to Mr. Marika Tuiwawa and Mr. Alivereti Naikatini for their assistance in the sample collection and deposition of samples in Fiji.

Additional Information and Declarations

Competing Interests

Author Contributions

Field Study Permissions

DNA Deposition

Data Availability

The authors declare that they have no competing interests.

Joape Ginigini performed the experiments, analyzed the data, contributed reagents/materials/analysis tools, prepared figures and/or tables, authored or reviewed drafts of the paper, approved the final draft.

Gaël J. Lecellier conceived and designed the experiments, performed the experiments, analyzed the data, contributed reagents/materials/analysis tools, prepared figures and/or tables, authored or reviewed drafts of the paper, approved the final draft.

Mael Nicolas performed the experiments, analyzed the data, contributed reagents/materials/analysis tools, authored or reviewed drafts of the paper, approved the final draft.

Mohammed Nour conceived and designed the experiments, authored or reviewed drafts of the paper, approved the final draft.

Edouard Hnawia contributed reagents/materials/analysis tools, authored or reviewed drafts of the paper, approved the final draft.

Nicolas Lebouvier conceived and designed the experiments, contributed reagents/materials/analysis tools, authored or reviewed drafts of the paper, approved the final draft.

Gaëtan Herbette performed the experiments, analyzed the data, contributed reagents/materials/analysis tools, prepared figures and/or tables, authored or reviewed drafts of the paper, approved the final draft.

Peter Lockhart analyzed the data, contributed reagents/materials/analysis tools, authored or reviewed drafts of the paper, approved the final draft.

Phila Raharivelomanana conceived and designed the experiments, performed the experiments, analyzed the data, contributed reagents/materials/analysis tools, authored or reviewed drafts of the paper, approved the final draft.

The following information was supplied relating to field study approvals (i.e., approving body and any reference numbers):

In New Caledonia, tamanu nuts were collected under the scientific authorization of the South Province No 2050-2014

The following information was supplied regarding the deposition of DNA sequences:

The accD data is available at NCBI (accession number 983209430) and psaA-ycf3 (accession number 983209444).

The following information was supplied regarding data availability:

The raw measurements are available in Datasets S1–S3. Each file provides the data for one sample from one geographic site. These data were used for the chemodiversity analysis to compare the profiles between the different regions.

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
