# Peer review of "Chemodiversity of Calophyllum inophyllum L. oil bioactive components related to their specific geographical distribution in the South Pacific region"

_PeerJ, doi:10.7717/peerj.6896_

## Round 0.1 · original submission · Major Revisions

· Academic Editor

Major Revisions

This manuscript presents new data on chemical and genetic variation in a geographically widespread plant species that has been used for medicinal and cosmetic purposes. Two novel chemicals are proposed and the reviewers require that NMR, MS and FTIR data for these two new chemicals be supplied as supplementary materials. I identified a number of concerns below regarding methodology / sample size (reviewers also requested clarification on methodology). There are some major issues with English language throughout the manuscript; following revision by the authors, the English should be carefully checked by an English expert prior to any re submission. Note that reviewer #3 supplied many comments as an attachment, which must be downloaded separately.

Specific comments

Introduction – A clear statement or objectives or hypothesis needs to be given at the end of the introduction. Line 44 says the aim is to identify genetic variation in DNA markers but this is not a clear statement of overall objectives of the study. I think the idea is to gain insights from integrating analysis of genetics and phenotype (a “polyphasic approach” referred to later in the discussion; I suggest defining this term in the introduction and citing other references that have advocated this approach); this would be a logical way to convey your aims / objectives, in my opinion.

Line 2-6 Beginning the manuscript with a string of 32 consecutive nouns will deter almost every potential reader I suggest beginning the Introduction with a broader opening paragraph about the value of combining studies of genetics and phenotypes (citing some general references about this). Then, you can introduce your study system; the scientific plant name is essential but it is not important to list every common name in a long string of nouns.

Line 63 “From the collected nuts sampled from French Polynesia, New Caledonia and Fiji, a batch of 5-6 nuts were picked and subjected to a small scale extraction” - this stated sampling methodology does not match the data reported in Figure 5. If a batch of 5-6 nuts were ground together from each region, then there should only be 3 data points shown on figure 5. If 5-6 nuts were separately analyzed from each region, there should be 15-18 points total on Figure 5. Methods need to clearly explain the sampling protocol.

Line 110 Leaf samples are referred to here but Table 2 refers to nuts. Are leaves from the same trees as nuts? How many of the samples listed in table 2 were extracted and how were they chosen? If not all of the samples were used, why are they listed in the table?

Line 127 What does “corrected for quality of data” mean? If data have been altered, that needs to be carefully justified.
Line 128 “initial sequence fragments were cleaned” - exactly what was done and why? Any alteration of data needs to be justified.

Line 244 – The meaning of “in house standards” is not clear. A standard should be purchased from a reputably chemical manufacturer with composition guaranteed by independent assay. “in house standards” are not suitable for use in peer-reviewed publications unless details can be given about how the standards were certified.

Line 279 “the most potent of all C. inophyllum compounds against HIV-1 cell’s” Cite reference here

Line 290 “we have looked at 3 localities as equivalent population.” What does this mean?
Line 292 standards from what?

Line 296-298 Although that data are not shown, my impression is that there is little or no chemical specificity among the three region. Instead, there are some differences in the area under each curve on the HPLC. This means there could be small differences in proportional representation of chemicals but no difference in the type of chemicals. This point should be discussed and wording should be clearly supported by data presented.

Line 305 I understand the point here but the wording is problematic. I think it should say “Chemodiversity”; It is possible fore chemodiversity to be “dictated by” genetic influence; however, it makes no sense to say that distribution is “dictated by” genetic influence. Rather, it is the reverse: genetic patterns could be “dictated by” distribution.

Line 306 “closed discriminatory fitness” What does it mean?

Line 314 “may have been caused by their evolutionary clocks” What does it mean?

Line 318 “variations occurring may have been caused by geographical differences and not completely through cryptic genetics differences “ What does it mean? In the discussion section, I am expecting to see the terms “plasticity” and / or “environmentally influenced” relating to a possible explanation of observed chemodiversity.

Line 329 Did these haplotypes have a different chemosignature in the HPLC? This should be checked.

Line 330 “Polyphasic taxonomy as seen here where both chemical diversity and genetic diversity have been used to delineate plant or organism across geographical boundaries appears to be an adequate method to document living matter.” - I don’t think “as seen here.. appears to be adequate” is supported because you were not able to combine genetic and phenotypic information to gain insights in this case. Wording should be reconsidered, or clarify why you think it has been adequate.

Line 335 This final sentence seems to be a random addition with no link to the previous discussion. I suggest a more integrated /carefully organized conclusion.

Table 2 – The data in this table do not match the sampling methodology given on line 60 of the Methods. The Methods refers to “A, B,C” but nothing similar appears in this table. The heading in the table need clarification. If “Nuts” = 4 and “Replicates per site” =3, I would assume this means 12 nuts total for Nouméa. However, following this logic, there would be hundreds of nuts total; whereas, line 167 says there are only 47 samples total. If different sets of samples were used for DNA versus chemical analysis, those samples should be clearly labeled in Table 2 I have no idea what the storage type “RNA Later” means.

Figure 4 – The caption refers to “The standard peaks” however my interpretation of this terminology is that known (purchased, certified) “standards” of each given chemical were passed through the HPLC to reveal the location of each standard peak. However, the methods do not discuss purchasing each of these chemicals and passing them through your HPLC to determine retention time of each standard chemical on your HPLC. The methods mentions purchasing callophylum oil but this does not allow direct determination of each component chemical on your HPLC. The wording of the caption should be revised to more clearly explain exactly how each (presumed) chemical was identified on the graphs. I am guessing that you compared peaks to previously published HPLC output for callophyllum oil Rather than referring to “chemical standards” in the caption, should it say “expected retention times for chemicals are based on previous analyses of Callophyllum oil by Laure et al., 2008 and Leu et al., 2009”?

Figure 5 – This is an important figure, however the information should be conveyed more clearly. The legend should use the same three named groups used in Table 2: Fiji, New Caledonia, and French Polynesia. At least tell us that diamonds are French Polynesia and group all of the diamond labels together in the legend. The color scheme currently seems to be random; presentation would be more effective if each geographic regions had a contrasting color range. Using related colors within each geographic region would allow more effective visualization of patterns.

Figure 6 “Uncharacterized compounds are mentioned Px.” English needs to be improved; current wording does not provides an adequate explanation.

Figure 7 – most of the samples listed in Table 2 are not shown in this figure. Why are so many samples missing from Figure 7 and how did you decide which samples to include?

Reviewer 1 ·

Basic reporting

The article should include sufficient introduction and background of accD and PsaA-Ycf3 to know the work better. Your introduction needs more detail I suggest that you improve the description at lines 40- 51 to provide more justification for your study.

Experimental design

Methods should be described with sufficient information to be reproducible by another investigator.
- Line 69-70: Why do you use EtOAc (not Hexane) in microscale extraction process?
- Line 84-96: Standard’s purification was not clear.
This fraction ……………. Yielding 12 fractions. Why were only fraction 7,9 and 11 submitted to repeated preparative HPLC?

Validity of the findings

Overall, this main idea of this study is not a high novelty, because previous study has investigated the genetic variations, likes the authors said in the introduction. The novelty of this study was the use of different marker.

Additional comments

This study aims to utilize barcoding universal gene markers in accD and PsaA-Ycf3 as tools to study the genetic variations occurring intra-geographically and inter-geographically from three South Pacific countries (French Polynesia, New-Caledonia and Fiji). Authors used tamanu oil for identifying new compounds and tamanu leaves for DNA analysis. This study can be published with revision and clarification of the issues.

·

Basic reporting

no comment

Experimental design

no comment

Validity of the findings

no comment

Additional comments

The manuscript is scientifically sound and it is written in clear sequences. It is supplied with appropriate and sufficient data.
The author is advised to justify/rectify a minor comments.

The author should emphasise that from which ground/data they can elucidate the isomer of Tamanolide E1 and E2.
Line 117, what type of gel? (SDS page/agarose?)
Line 283, it should be Fig 4.
Line 199, use ring instead of cycle.
To differentiate the structure of Tamanolide E1 and E2 in Fig 3.
Caption in Table 3, please use symbol instead of parenthesis.

Reviewer 3 ·

Basic reporting

No comment
(Detail report is in the attached file)

Experimental design

No comment
(Detail report is in the attached file)

Validity of the findings

No comment
(Detail report is in the attached file)

Additional comments

Overall, the manuscript is generally well-written. The backgroud of the study was clear and the experiments were well-design with satisfied parameters.
There are corrections that can be seen in attached file.
Since authors proposed new compounds, authors should provide the NMR, IR and MS data of those new compounds in supplementary files.

Annotated reviews are not available for download in order to protect the identity of reviewers who chose to remain anonymous.

---

## Round 0.2 · Major Revisions

· Academic Editor

Major Revisions

The reviewers have requested some minor clarifications in the text. However, the main issue remaining is English language. The text contains many grammatical errors and awkward English sentences. I found the Abstract especially difficult to follow. Thus, the text requires major revisions of English. The references also need consistent formatting. I have edited the English using "track changes" in Word. I also typed several comments on the text that should be addressed by the authors during revisions. The editorial system only allows me to upload a PDF of the edited manuscript (which I have done); however, I will ask if the journal staff can forward the Word file because it is easier for authors to work with, and my comments were not preserved on the PDF file that I uploaded.

Reviewer 1 ·

Basic reporting

It is Clear and sufficient field background.

Experimental design

Please put material information, such the solvent, the reagent, etc.

Validity of the findings

Conclusions are well stated.

Additional comments

This study can be published with minor revisions

·

Basic reporting

no comment

Experimental design

Line 154: Samples were immersed in EtOAc and sonicated for 5 mins before this was
155 partitioned with EtOH at a 1:1 ratio v/v....

The sentence is not clear. Which one is the first?
Partitioning between which solvent?....EtOAc and EtOH are miscible. The authors need to clarify this method.

Validity of the findings

no comment

Reviewer 3 ·

Basic reporting

This the second review. The improvement made by the authors are adequate and satisfactory. I have no further comment on this manuscript.

Experimental design

This the second review. The improvement made by the authors are adequate and satisfactory. I have no further comment on this section.

Validity of the findings

This the second review. The improvement made by the authors are adequate and satisfactory. I have no further comment.

---

## Round 0.3 · accepted · Accept

· Academic Editor

Accept

The authors have made appropriate corrections. I have just a few small edits for the final submission:
L 23 I believe it should say “efficacy” rather than “efficiency”
L 78 I believe it should say “efficacy” rather than “efficiency”
L 165 “280 nm, wavelength” to “280 nm, the wavelength”
L 166 “The removal of noise,” to “To remove noise,”
L 257 “ \Microsoft” to “ Microsoft”
L357 “two close set” to “two close sets”

#